# Machine Learning Based Analysis of Relations between Antigen Expression and Genetic Aberrations in Childhood B-Cell Precursor Acute Lymphoblastic Leukaemia

**DOI:** 10.3390/jcm11092281

**Published:** 2022-04-19

**Authors:** Jan Kulis, Łukasz Wawrowski, Łukasz Sędek, Łukasz Wróbel, Łukasz Słota, Vincent H. J. van der Velden, Tomasz Szczepański, Marek Sikora

**Affiliations:** 1Department of Pediatric Hematology and Oncology, Medical University of Silesia in Katowice, ul. 3 Maja 13-15, 41-800 Zabrze, Poland; jkulis@sum.edu.pl (J.K.); slota.lukasz2392@gmail.com (Ł.S.); tszczepanski@sum.edu.pl (T.S.); 2Łukasiewicz Research Network—Institute of Innovative Technologies EMAG, 40-189 Katowice, Poland; lukasz.wawrowski@emag.lukasiewicz.gov.pl; 3Department of Microbiology and Immunology, Medical University of Silesia in Katowice, ul. Jordana 19, 41-808 Zabrze, Poland; lsedek@sum.edu.pl; 4Department of Computer Networks and Systems, Silesian University of Technology, 44-100 Gliwice, Poland; lukasz.wrobel@polsl.pl; 5Department of Immunology, Erasmus MC, University Medical Center Rotterdam, 3015 GD Rotterdam, The Netherlands; v.h.j.vandervelden@erasmusmc.nl

**Keywords:** acute lymphoblastic leukaemia, flow cytometry, cytogenetics, machine learning, knowledge discovery, classification

## Abstract

Flow cytometry technique (FC) is a standard diagnostic tool for diagnostics of B-cell precursor acute lymphoblastic leukemia (BCP-ALL) assessing the immunophenotype of blast cells. BCP-ALL is often associated with underlying genetic aberrations, that have evidenced prognostic significance and can impact the disease outcome. Since the determination of patient prognosis is already important at the initial phase of BCP-ALL diagnostics, we aimed to reveal specific genetic aberrations by finding specific multiple antigen expression patterns with FC immunophenotyping. The FC immunophenotype data were analysed using machine learning methods (gradient boosting, decision trees, classification rules). The obtained results were verified with the use of repeated cross-validation. The t(12;21)/ETV6-RUNX1 aberration occurs more often when blasts present high expression of CD10, CD38, low CD34, CD45 and specific low expression of CD81. The t(v;11q23)/KMT2A is associated with positive NG2 expression and low CD10, CD34, TdT and CD24. Hyperdiploidy is associated with CD123, CD66c and CD34 expression on blast cells. In turn, high expression of CD81, low expression of CD45, CD22 and lack of CD123 and NG2 indicates that none of the studied aberrations is present. Detecting aberrations in pediatric BCP-ALL, based on the expression of multiple markers, can be done with decent efficiency.

## 1. Introduction

Acute lymphoblastic leukaemia (ALL) is the most frequent leukaemia in children. There are two main types of ALL, one of them is B-cell precursor acute lymphoblastic leukaemia (BCP-ALL), comprising about 85% of all ALL cases. Abnormal leukemic cells (blasts) can be distinguished by assessing the immunophenotype i.e., the composition of specific antigens expressed on the cell surface or inside the cytoplasm with the use of flow cytometry technique (FC). There are several genetic aberrations related with BCP-ALL that have evidenced prognostic significance. Most common are translocations t(12;21), t(9;22), t(1;19) and translocations within the q23 region of chromosome 11 (t(11q23)), resulting in generation of fusion genes ETV6-RUNX1, BCR-ABL1, TCF3-PBX1 and KMT2A gene rearrangements, respectively [1,2]. Other types of aberrations manifest as increase or decrease in chromosome number corresponding to hyper- and hypodiploidy, respectively [3].

Genetic tests are generally considered expensive and have a long turnaround time from sample collection till result generation. On the contrary, FC technique appears to be more broadly available, affordable and the results can be obtained on the same working day.

It is already known that expression, or lack of certain antigens is associated with molecular aberrations. For instance, expression of NG2 and CD15 is associated with t(v;11q23)/KMT2A gene rearrangements [1,4,5,6] and high expression of CD123 is often seen in patients with hyperdiploidy [3,7,8]. FC results are most commonly evaluated with the use of 2-dimensional dot plots representing expression pattern of 2 antigens at a time (Figure 1 and Figure 2). Each dot plot shows a pair of antigens assessed in patients with BCP-ALL. Coloured dot populations represent individual patients with particular genetic aberration: red—t(v;11q23)/KMT2A; green—t(12;21)/ETV6-RUNX1 and blue—hyperdiploidy. The dot in the middle of each population represents the median of fluorescence intensity (MFI). One can see that, this ’classical’ approach of immunophenotype assessment is not sufficient to infer the underlying genetic aberrations, even if the above mentioned phenotypic features are present.

To date, only a few papers describing potential association of quantitatively measured antigen expression with genetic aberrations exist in the literature [2,4]. However, this previous research covered only one-dimensional statistical analysis and the aberration prediction efficiency was examined only on the training set of examples. In addition, the authors did not make the analysed data sets available. Therefore, the results reported in these papers might not be optimally estimated.

This article presents an original approach of finding multiple (multi-dimensional) antigen expression patterns revealing possible genetic abnormalities underlying individual BCP-ALL cases. Our innovative approach exploits quantitative antigen expression measure and advanced multidimensional data analysis, including cross-validation and releasing the Shiny application for researchers to predict the occurrence of particular aberrations based on the expression of multiple markers by FC. As genetic aberrations have evidenced prognostic significance, success in finding immunophenotype-genotype correlation would allow to determine the patient prognosis already at the initial stage of diagnosis with FC immunophenotyping. The results show also the real specificity, positive and negative predictive values of the predictive models.

Machine learning methods, particularly data clustering [9,10,11] and outlier detection [12] were also previously used to visualize and interpret flow and mass cytometry data. These papers concern the stage of raw data processing.

## 2. Materials and Methods

In this study, we assessed the expression of surface markers CD34, CD45, CD10, CD38, CD20, CD33, CD13, CD22, CD24, CD9, CD15 + CD65, CD66c, CD123, CD81, NG2, and cytoplasmic cyIgM and TdT. The study group consisted of 818 pediatric patients (Median age 4 years (range 0–17); sex ratio M/F = 1.17). All patients were diagnosed in major Polish pediatric hematooncology centers in years 2008–2015. The EuroFlow BCP-ALL diagnostic panel and standardized operating protocols for sample preparation were applied using the eight-colour FC technique [13,14].

Cell analysis and gating the leukemic blasts out of all events followed the exclusion of doublets and cellular debris based on the forward and side scatter parameters. Then, the expression of antigens on blasts was measured as median of fluorescence intensity (MFI). To normalize the antigen expression, raw MFI values were transformed into scores of a normalized (nMFI) scale, as reported in [15]. For each of the studied antigens, a negative and positive reference population (observable in healthy donor samples) was chosen. For example, for CD66c the reference positive population was neutrophils (because it has the highest expression of CD66c), while the negative population was erythroblasts (because it never expresses this antigen). The ranges of MFI between those cell types were divided into 10 equal antigen-specific steps (scores). Expression level corresponding to the negative control cell population was given 0 nMFI score, while 10 nMFI equalled the expression of positive reference cell population which in classical notation corresponds to bright expression. The expression (measured as MFI) of particular antigen on leukemic blasts was then compared against the nMFI scale obtained for normal cells and ascribed respective nMFI score. Therefore, expression level of 4–6 in classical notation could be described as “moderate” or “dim”, while nMFI = 1 is considered low-positive or weak. Determination of the nMFI step width, enabled to extrapolate the scale over the normal reference MFI values, making possible to assess marker overexpression on leukemic blasts (nMFI > 10) which in classical notation is still defined as bright expression. The use of here proposed normalized scale of antigen expression provides more information than classical approach with the use descriptive notations which are often arbitrary. Gradual antigen expression assessment was chosen in order to associate the genetic abnormalities with particular levels of antigen(s) expression level determined quantitatively. For markers without identifiable positive control (e.g., NG2), binary type of expression was used

The total raw data set contained 818 patients and 21 parameters—17 antigens with measured expression levels and 4 genetic aberrations (binary type attributes). Each decision attribute described one decision (classification) problem. Given the generally exclusive nature of genetic aberrations studied here, the paper formulated five decision problems:t(12;21)/ETV6-RUNX1 translocation prediction (in the analysed data set the primary decision class indicates patients with this translocation),t(v;11q23)/KMT2A translocation prediction (in the analysed data set the primary decision class indicates patients with this translocation),hyperdiploidy prediction, (although hyperdiploidy can coexist with other genetic aberrations, in our dataset there were only 4 such examples; of in the analysed data set the primary decision class indicates patients with this aberration),no aberration (in the analysed data set the primary decision class indicates patients with no one of the above mentioned aberrations),t(12;21)/ETV6-RUNX1 vs. hyperdiploidy (the problem concerns the possibility of discrimination between t(12;21)/ETV6-RUNX1 and hyperdiploidy; the primary class indicates patients with t(12;21)/ETV6-RUNX1 translocation while the secondary class indicates patients with hyperdiploidy); examples with neither t(12;21)/ETV6-RUNX1 nor hyperdiploidy were removed from the analysis; in the data set there were only 4 examples with both t(12;21)/ETV6-RUNX1 and hyperdiploidy these examples were also removed from the analysis.

Initially, the dataset included other important recurrent genetic aberrations - BCR-ABL1, and TCF3-PBX1. However, there were only 7 cases of BCR-ABL and 6 of TCF3-PBX with many missing results. Therefore, these aberrations were discarded from the analysis.

During analysis each decision problem was considered separately. For 8 out of 17 antigens with measured expression levels the number of missing values exceeded 20% with maximum equal to 36% for CD13 (see Missing data section in the Appendix A). For 6% observations the concurrent prevalence of missing values for attributes CD66c, cyIgM, CD33, CD13, CD22, CD9, CD81 was observed. Since the complete vector of antigen expression was not available for every case, the data set had to be preprocessed. In the first stage of preprocessing, cases with 50% or more missing values were removed. During the further analysis no imputation of missing values was carried out. Table 1 presents the descriptive statistics of conditional attributes. Table 2 shows detailed information about the data sets. One can notice that the data sets are imbalanced.

The study was conducted from two perspectives: predictive and descriptive (explanatory). The purpose of the predictive analysis was to build classifiers resolving above-mentioned decision problems. The aim of the explanatory analysis was to find important dependencies between decision attribute (e.g., aberration) and conditional attributes (i.e., antigen expression). Figure 3 presents the detailed workflow of the analysis.

### 2.1. Predictive Analysis

Predictive models were built using Gradient Boosting Machine (GBM) [16] method implemented in h2o R package. The main idea of GBM is inducing an ensemble of shallow trees in sequence with each tree learning and improving on the previous one. The decision of selection of GBM in this study has been made after running *automl* procedure [17] available in h2o R package. The GBM achieved the best prediction efficiency among other methods examined by *automl* i.e.,: Distributed Random Forest, Generalized Linear Model, Naïve Bayes Classifier and Support Vector Machine. The comparison of considered algorithms is available in Appendix A. During the experiments stratified 5-fold cross validation procedure repeated 10 times was applied. Thus, for each decision problem 50 classification models were built. As the loss function the harmonic mean of precision (*PPV* —Positive Predictive Value) and sensitivity (*TPR*—True Positive Rate) known as $F_{1}$ score (Equation 1) was used.

Additionally, in the literature of the subject there are also empirical research showing that GBM outperforms other classification methods [18].
(1)$$F_{1}=\frac{2\cdot PPV\cdot TPR}{PPV+TPR}$$
where *PPV* = $\frac{TP}{TP+FP}$, *TPR* = $\frac{TP}{TP+FN}$ and TP (True Positives), TN (True Negatives), FP (False Positives), FN (False Negatives) are elements of a classifier confusion matrix.

Additionally variable importance was calculated using model-specific measure and model-agnostic approach based on permutation test with DALEX R package [19]. Moreover, in Shiny application, we used Shapley value [20] to assess the impact of feature in final prediction.

### 2.2. Descriptive Analysis

The main aim of the descriptive analysis was to discover multidimensional relations between antigen expression and molecular aberrations. Two strategies to find antigen-aberration dependencies were applied: divide-and-conquer and separate-and-conquer. Utilise of two approaches allow on finding such relationships which will be impossible to obtain using only one of mentioned technique. The firs strategy utilises RPART decision tree induction algorithm [21], while the second one uses RuleKit, the decision rule induction package [22].

The descriptive analysis was carried out in two paths. The first path involves RPART decision tree induction, the second one involves RuleKit-based rule induction. Descriptive analysis was carried out on the entire set of examples. To avoid the data over-fitting and get decision tree with reasonable number of levels during the experiments the process of decision tree growing was stopped when Cohen‘s Kappa statistic (Equation 2) [23] calculated on the entire set of examples attains value 0.8. Kappa statistics is a measure of accuracy agreement between two classifiers, the evaluated classifier (ACC) and the random classifier which classifiers examples in accordance with class distribution (RAND). Kappa statistic is a special case of association that indicates an “association” of the elements of a classifier confusion matrix on its main diagonal.
(2)$$\kappa =\frac{\mathrm{ACC}-\mathrm{RAND}}{1-\mathrm{RAND}}$$
where
$$\mathrm{ACC}=\frac{TP+TN}{TP+TN+FP+FN},$$
$$\mathrm{RAND}=\frac{\left(TN+FP)(TN+FN)+(FN+TP)(FP+TP\right)}{{\left(TP+TN+FP+FN\right)}^{2}}.$$

In rule-based descriptive analysis, the induced rules can overlap each other. Each induced rule can be viewed as a simple classier working on limited set of examples - the examples covered by the rule. Each rule *r* has the form:


$$\;\mathrm{IF}\;w_{1}\mathrm{and}w_{2}\mathrm{and}\ldots \mathrm{and}w_{n}\mathrm{THEN}C$$


The premise of a rule is a conjunction of elementary conditions $w_{i}\equiv a_{i}⊙x_{i}$, with $x_{i}$ being an element of the $a_{j}$ domain and ⊙ representing a relation (=for symbolic attributes; <,≤,>,≥ for ordinal and numerical ones). The value *C* in the rule conclusion indicates one of the decision class identifier. The meaning of the rule is as follows: if an examples fulfils all conditions specified in the conditional part of the rule, then it belongs to the decision class specified in the rule conclusion. Additionally, an example satisfying the conditions specified in the rule premise is stated to be covered by the rule. The examples whose labels are the same as the conclusion of *r* are called positive examples, while the others are called negative examples.

In our research, as the rule search heuristic in rule induction algorithm [22], the C2 (Equation 3) rule quality measure was used. During the induction, the form of a rule premise was optimised towards the C2 value maximisation.
(3)$$\left(\frac{Np-Pn}{N(p+n)}\right)\left(\frac{P+p}{2P}\right).$$
where *P* is the number of examples from the primary class, *N* is the number of examples from the secondary, *p* is the number of examples form the primary class satisfying the conditions in rule premise (i.e., positive examples covered by the induced rule), *n* is the number of examples form the secondary class satisfying the conditions in rule premise (i.e., negative examples covered by the induced rule).

The C2 measure can be viewed as the weighted version of Kappa statistics. *C*2 evaluates each rule separately. During the rule induction, the form of a rule premise was optimised towards the *C*2 value maximisation.

For rule-based exploratory analysis the rule precision (p/(p+n)) rule coverage (p/P) and rule statistical significance (Fisher’s exact test with *p*-value adjustment) were also reported.

## 3. Results

### 3.1. Predictive Analysis

Figure 4 shows ROC curves—with 95% confidence intervals—for all considered decision problem. ROC curves were calculated on the independent test data sets. Table 3 presents values of sensitivity, specificity, PPV and NPV for all decision problems. In parenthesis standard deviations are presented.

The overall classification accuracy of GBM classifier varied from 71% for no aberration to 98% for t(v;11q23)/KMT2A aberration. $F_{1}$ score takes values from 0.52 (t(12;21)/ETV6-RUNX1) to 0.81 (t(12;21)/ETV6-RUNX1 vs. hyperdiploidy).

After model training the feature importance analysis was carried out. GBM algorithm allows to generate a feature importance ranking. The final feature importance was calculated as the average position of the feature importance in 50 classifiers (as was mentioned, experiments were carried out in 10 × 5CV mode).

For t(12;21)/ETV6-RUNX1 aberration antigens CD10, CD9, CD81, CD45, CD22, CD66 and CD34 were identified as the most important features. For t(v;11q23)/KMT2A the following six most important features were identified: CD45, TdT, CD24, CD10, cyIgM, CD34. It is interesting that the NG2 feature was only ranked 7th in the feature importance ranking (see Variable importance section in the Appendix A). The crucial feature in hyperdiploidy prediction is CD123 — it was on first place of feature importance in all trials of cross-validation. On subsequent places there were CD66, CD24, CD34, CD10 and CD9. In the case of no aberration prediction the following feature importance ranking was obtained: CD123, CD10, CD9, CD24, CD22, CD34, and CD45. For t(12;21)/ETV6-RUNX1 vs. hyperdiploidy prediction the most important features were: CD123, CD66c, CD34, CD10, CD9, CD22, and CD81.

The above results were obtained for classifiers trained on all available features. The purpose of the next experiment was to check whether is it possible to achieve similar classification accuracy with limited set of features. For this purpose, the simple strategy was applied. Basing on the feature importance ranking of all features, classifiers with iteratively extended feature set were trained. The first classifier worked only with the best feature, the second with two top-ranked features and so on. Consistently, all calculations were conducted in 10x5CV mode. The results of this experiment are presented in Figure 5.

For all decision/prediction problems one can observe that the $F_{1}$ value grows with the number of variables used in prediction model. In the vast majority of analysed cases the difference between the classifiers is not significant. It it difficult to indicate optimal number of features in classifier; however, one can observe that the $F_{1}$ value stabilises when the six most important features are used in the prediction.

### 3.2. Descriptive Analysis

#### 3.2.1. Decision Trees

For each of the considered decision problem the RPART [24] decision tree was induced based on the entire available data set. The grid based hyper-parameter optimisation procedure was used to find the trees with high explanation efficiency. To get precise and transparent decision trees the induction process was stopped when during tree induction the Kappa coefficient exceeded value 0.8. Table 4 presents values of classification measures of induced trees.

Induced trees characterised different complexity. The number of tree levels varied form 5 (ETV6-RUNX1) to 12 (no aberration), the number of leaves varied for 9 (ETV6-RUNX1) to 81 (no aberration). Based on the induced decision trees the feature importance rankings were determined. The ranking of the six most important features for each decision problem is as follows:ETV6-RUNX1: CD10, CD24, CD34, CD38, CD45, TdT;KMT2A: NG2, CD24, cyIgM, CD34, CD123, CD10;hyperdiploidy: CD123, CD24, CD10, CD34, CD9, TdT;no aberration: CD10, Cd24, CD34, TdT, CD123, CD66c;ETV6-RUNX1 vs. hyperdiploidy: CD66c, CD10, CD24, TdT, CD123, CD34.

In the Appendix A, plots and feature rankings of all induced decision trees are presented. Below, the strongest—covered by the maximal number of examples—rules for each of the induced decision trees are presented. Each rule represents the decision path form a root of the considered tree to its leaf. For each decision tree two rules that cover the largest number of examples are presented. The first rule—positive rule—describes the set of examples with considered aberration, the other—negative rule—rule describes examples without aberration.

For t(12;21)/ETV6-RUNX1 aberration the following positive and negative rules were induced:1.IF CD66c = 0 and CD45 < 2 and CD24 ∈ [25,38) and CD34 < 4 THEN ETV6-RUNX1;2.IF CD66c ≥ 1 and CD10 < 98 and CD13 < 24 and CD38 < 20 and CD24 < 101 and CD10 < 25 and CD9 ≥ 2 THEN no ETV6-RUNX1;

The first rule covers 18 examples, 17 examples represent patients with ETV6-RUNX1 aberration. The rule covers 32% out of all patients with ETV6-RUNX1. The second rule covers 127 examples, all examples represent patients with no ETV6-RUNX1 aberration (30% of patients with no ETV6-RUNX1). Figure 6 (t(12;21)/ETV6-RUNX1) presents the increasing values of *Odds ratio* for the premise of the positive rule (1) with successive elementary conditions added. For elementary condition (CD66c = 0) the *Odds ratio* equals 4.47. It means that the patients with CD66c expression equal to 0 are above four times more likely to have ETV6-RUNX1 aberration that patents with CD66c ≥ 1. For rule with two elementary conditions (CD66c ≥ 1 and CD10 < 98) the *Odds ratio* equals 8.37. For the whole positive rule *Odds ratio* equals 16.90.

For t(v;11q23)/KMT2A aberration the following positive and negative rules were induced:3.IF NG2 = 1 and CD24 < 5 THEN KMT2A;4.IF NG2 = 0 and CD123 < 9 and cyIgM < 18 and CD20 < 9 and CD15 + CD65 = 0 THEN no KMT2A;

The first rule (3) covers six examples, all examples represent patients with KMT2A aberration. The rule covers 35% out of all patients with KMT2A. The second rule (4) covers 580 examples, only three examples represent patients with KMT2A aberration. The rule covers 98% of examples representing the “no KMT2A” decision class. The whole premise of the positive rule (NG2 = 1 and CD24 < 5) covers no negative examples, that is why the *Odds ratio* is undefined (but it goes to infinity). When only one elementary condition (NG2 = 1) is considered the *Odds ratio* value for the positive rule equals 219.3.

For hyperdiploidy the following positive and negative rules were induced:5.IF CD123 ≥ 1 and CD9 ≥ 3 and CD24 ≥ 14 and CD33 ≥ 1 and CD20 ≥ 2 and CD34 ≥ 12 THEN hyperdiploidy;6.IF CD123 = 0 and CD13 < 16 and CD81 < 13 and CD66c = 0 and TdT < 9 THEN no hyperdiploidy;

The first rule (5) covers 25 examples, 22 examples represent patients with hyperdiploidy. The rule covers 18% out of all patients with hyperdiploidy. The second rules (6) covers 196 examples and 16 out of them represent patients with hyperdiploidy. The rule covers 39% of examples representing the “no hyperdiploidy” decision class. Figure 6 (hyperdiploidy) presents the increasing values of *Odds ratio* for for the premise of the positive rule (5) with successive elementary conditions added.

For no aberration decision problem the following positive (there exists an aberration) and negative (no aberration) rules were induced:7.IF CD123 ≥ 1 and CD45 < 4 and CD9 ≥ 3 and CD24 ∈ [15,84) THEN there exists aberration;8.IF CD123 = 0 and NG2 = 0 and CD45 ≥ 2 and CD22 ≥ 3 and CD81 ≥ 12 THEN no aberration;

The first rule (7) covers 89 examples, 68 examples represent patients with any aberration. The rule covers 35% out of all patients with any aberration. The second rule (8) covers 45 examples and four out of them represent patients with any aberration. The rule covers 17% of examples representing the “no aberration” decision class. Figure 6 (no aberration) presents the increasing values of *Odds ratio* for the premise of the positive rule (7) with successive elementary conditions added.

For t(12;21)/ETV6-RUNX1 vs. hyperdiploidy decision problem the following positive (t(12;21)/ETV6-RUNX1) and negative (hyperdiploidy) rules were induced:9.IF CD66c = 0 and CD10 ≥ 31 and CD13 < 14 and CD38 ≥ 1 THEN ETV6-RUNX1 and no hyperdiploidy;10.IF CD66c ≥ 1 and CD123 ≥ 1 and CD10 < 98 and CD24 < 36 and CD45 < 4 THEN hyperdiploidy and no ETV6-RUNX1;

The first rule (9) covers 21 examples, 20 examples represent patients with ETV6-RUNX1 and no hyperdiploidy. The rule covers 38% out of all patients with ETV6-RUNX1. The second rule (10) covers 76 examples; only three out of them represent patients with ETV6-RUNX1. The rule covers 63% of examples representing the “hyperdiploidy and no ETV6-RUNX1” decision class. Figure 6 (t(12;21)/ETV6-RUNX1 vs. hyperdiploidy) presents the increasing values of *Odds ratio* for the premise of the rule (9) with successive elementary conditions added. In addition, Figure 6 (hyperdiploidy t(12;21)/ETV6-RUNX1) presents the increasing values of *Odds ratio* for the premise of the rule (10).

#### 3.2.2. Multi-Attribute Decision Rules

A sequential covering approach to the rule induction allowed to determine a few additional interesting rules. Below, there are two rules characterising high values of rule support and rule precision. The rules are also statistically significant

The first rule has a form of:11.IF CD9 > 1 and CD13 < 3 THEN no ETV6-RUNX1;

The rule is covered by 170 positive examples—which stands for 40% of all examples from the “no ETV6-RUNX1” decision class—and only one negative example representing a patient with the ETV6-RUNX1 aberration. The statistical significance of this rule equals $5.0\times 10^{-10}$, while the *Odds ratio*—34.4. The rule with only one (CD9 > 1) elementary condition covers 216 examples—210 examples form “no ETV6-RUNX1” class and 6 examples form ”ETV6-RUNX1” decision class.

Logical negation of the above rule takes a form IF CD9 ≤ 1 OR CD13 ≥ 3 THEN ETV6-RUNX1. This rule covers 126 examples (24 with ETV6-RUNX1 aberration and 102 without this aberration). The rule covers 46% of examples representing “ETV6-RUNX1” decision class, but this rule has low precision and *Odds ratio* equals 2.69 only.

The second rule has a form of:12.IF TdT > 2 and CD45 < 3 THEN no KMT2A;

The rule is covered by 405 positive examples—which stands for 69% of all examples from the “no KMT2A” class—and only one negative example representing a patient with KMT2A aberration. The statistical significance of this rule equals $1.5\times 10^{-7}$, while the *Odds ratio*—35.4. The rule with only one (TdT > 2) elementary condition covers 481 examples—476 examples from “no KMT2A” decision class and five examples representing “KMT2A” class. It is worth to noticing that five examples constitute 29% examples of the whole “KMT2A” decision class.

The above two rules were generated automatically. The next reported rules were induced within the user-driven mode to verify certain hypotheses about the co-occurrence of the considered aberrations.

#### 3.2.3. Aberration t(12;21)/ETV6-RUNX1

In the first experiment, it was enforced that the elementary condition (CD9 > 1) has to occur in the rule premise. Two rules were obtained that covered a large group of examples representing patients without the ETV6-RUNX1 aberration:13.IF CD9 > 1 and CD13 < 24 and CD33 = 0 THEN no ETV6-RUNX1;

The rule is covered by 181 positive examples—43% of the whole decision class—and 2 negative examples. The statistical significance of the rule is $1.1\times 10^{-9}$, while the *Odds ratio*—18.78.

14.IF CD9 > 1 and CD10 < 106 THEN no ETV6-RUNX1;

The rule is covered by 205 positive examples—43% of the whole decision class—and 5 negative examples. The statistical significance of the rule is $1.3\times 10^{-8}$, while the *Odds ratio*—8.88.

In the second experiment, it was checked whether the CD81 antigen is a prognostic factor in the ETV6-RUNX1 aberration. The occurrence of only this antigen was enforced in the premises of rules indicating this decision class. The following rule was achieved:15.IF CD81 ≤ 4 THEN ETV6-RUNX1;

The rule is covered by 40 positive examples—which stands for 77% of all examples from the “ETV6-RUNX1” class, but—unfortunately—also 250 examples from the class indicating the lack of this aberration. The *Odds ratio* for this rule is only 2.29.

Due to the above, in the successive experiment the occurrence of the (CD81 ≤ 4) condition was enforced. This time is was also allowed to add other elementary conditions to the rule premise so that the rule cover fewer negative (without ETV6-RUNX1 aberration) examples. The following rules were induced:16.IF CD81 ≤ 4 and CD38 ≥ 3 and CD10 ∈ [16,58] and CD66c = 0 and CD22 ≤ 2 THEN ETV6-RUNX1;17.IF CD81 ≤ 4 and TdT ≤ 8 and CD24 ≤ 23 and CD10 ≥ 47 and CD66c = 0 and CD22 ≤ 2 THEN ETV6-RUNX1;

The first rule covers five positive examples, and the second one, seven positive examples. Neither of the rules covers negative examples. The statistical significance of the rules is $1.3\times 10^{-5}$ and $1.3\times 10^{-7}$, respectively. Both rules are quite specific—they have many elementary conditions and each rule covers only 10% of examples from the “ETV6-RUNX1” class.

#### 3.2.4. Aberration t(v;11q23)/KMT2A

In the case of the t(v;11q23)/KMT2A aberration we searched for the rules which described the “KMT2A” decision class and contained at least one of CD45, CD24 antigens. The following three strong rules were found:18.IF NG2 = 1 and CD24 ≤ 4 THEN KMT2A;

The rule is covered by eight positive examples—47% of patients with the KMT2A aberration—and no negative examples. The statistical significance of the rule is $5.7\times 10^{-14}$.

19.IF CD45 ≥ 4 and CD24 ≤ 5 and NG2 = 1 THEN KMT2A;

The rule is covered by seven positive examples—41% of patients with the KMT2A aberration—and no negative examples. The statistical significance of the rule is $3.4\times 10^{-12}$. After removing the elementary condition (NG2 = 1) from the rule premise the rule still covers 7 positive examples but also 6 negative examples (*Odds ratio* equals 67.9)

20.IF CD24 ≤ 5 and CD45 ≥ 3 and cyIgM ≤ 3 and NG2 = 1 THEN KMT2A;

The rule is covered by eight positive examples—47% of patients with the KMT2A aberration—and no negative examples. The statistical significance of the rule is $5.7\times 10^{-14}$. After removing the elementary condition (NG2 = 1) from the rule premise the rule still covers 8 positive examples but also 13 negative examples (*Odds ratio* equals 39.31).

#### 3.2.5. Hyperdiploidy

In the user-driven rule induction for the hyperdiploidy decision problem it was checked whether the CD123 antigen is enough to precise distinguish patients with and without hyperdiploidy. For this purpose, there two rules:21.IF CD123 = 0 THEN no hyperdiploidy;22.IF CD123 ≥ 1 THEN hyperdiploidy;

—were manually defined and based on a such defined classification system all examples were classified. The classification accuracy for that classifier is very low—the classifier is able to correctly classify only 30% examples from each decision class. The first rule covers only 19 positive examples and 210 negative ones. The second rule covers 84 positive examples and 77 negative ones. In both cases the rules are not statistically significant.

In the next experiment then, the user-driven rule induction was launched. It was enforced that at least the CD123 attribute has to occur in the rule premise. Three interesting rules indication the “hyperdiploidy” decision class were induced:23.IF CD123 ≥ 1 and CD20 < 2 and TdT < 5 and CD10 ∈ (6,20) THEN hyperdiploidy;

The rule is covered by 20 positive examples—16% of patients with hyperdiploidy—and 5 negative examples. The statistical significance of the rule is $1.8\times 10^{-10}$, while the *Odds ratio*—14.35.

24.IF CD123 ≥ 4 and CD34 ≥ 6 and CD24 ≥ 13 and CD10 ≤ 67 THEN hyperdiploidy;

The rule is covered by 10 positive examples—8% of patients with hyperdiploidy—and no negative examples. The statistical significance of the rule is $6.4\times 10^{-7}$.

25.IF CD123 ≥ 3 and CD34 ≥ 12 and CD10 ≤ 80 THEN hyperdiploidy;

The rule is covered by 11 positive examples—9% of patients with hyperdiploidy—and no negative examples. The statistical significance of the rule is $1.5\times 10^{-7}$.

## 4. Discussion

Since the ‘classical’ approach of FC data analysis based on 2-dimensional dot plots examination is not sufficient to detect possible underlying genetic aberrations, we employed advanced analysis tools (including RPART decision tree induction and Gradient Boosting Machines) to find correlations between the entire immunophenotype of leukemic blasts, characterized quantitatively with 17 markers and genetic abnormalities of patients with BCP-ALL. We focused on identification of the three most frequent genetic abnormalities: ETV6-RUNX1, KMT2A rearrangements, hyperdiploidy and none of the mentioned aberration status. We also tried to identify specific differences in expression of the studied markers to discriminate between ETV6-RUNX1 and hyperdiploidy.

In the majority of previous studies in this field, expression of markers in terms of correlation with genetic results was described only qualitatively (as positive or negative) [3,6,25]. Different approach was proposed by Tsagarakis et al. [2] who used an original scoring system to estimate the chance of given aberration to occur based on expression of antigens presented as percentage of blasts positive for given marker.

We proposed to use a normalized (graded) scale of antigen expression based on median fluorescence intensity. Gradual antigen expression scale enabled to investigate for possible genetic abnormalities associated with certain level of expression of several markers simultaneously. We showed that, indeed, high expression levels of given markers is associated with occurrence of particular aberration, while moderate or low can indicate otherwise. Furthermore, also on the statistical field, modern and innovative approach was applied. Combination of the above allowed to find specific antigenic patterns, with gradually assessed expression, strongly associated with aberrations.

### 4.1. t(12;21)/ETV6-RUNX1

We have come to the conclusion that CD10 is the most important marker in both predictive and explanatory analysis and showed that very high expression of CD10 correlates with ETV6-RUNX1 occurrence, while moderate and low expression of CD10 is more frequent in patients without this aberration. De Zen et al. also previously proposed that CD10 increases probability of ETV6-RUNX1 [26].

Low expression of CD34 and CD45 was also important in both predictive and explanatory analysis. This was already observed by Tsagarakis et al. [2], who proposed that weak expression of CD34 (along with other conditions—see full rules in results section) is strongly associated with ETV6-RUNX1. De Zen et al. [26] have also proposed that the lack of CD45 and CD34 increases the chance of ETV6-RUNX1. On the other hand, Borowitz et al. suggested that this translocation might be related to high expression of CD45 however the specificity of their results was low [27]. Hrusak et al. observed that lack of CD66c antigen on blast cells is more frequent in patients with t(12;21) which can also be seen in our results as the expression of CD66c marker was one of the most important antigens associated with ETV6-RUNX1 in predictive analysis [28]. Our research also confirms previous findings that expression of CD24 is associated with ETV6-RUNX1 [3], however, the use of gradual expression scale pointed out more precisely that moderate expression of CD24 on blast cells has the strongest correlation. For the first time, we have demonstrated that low expression of CD38 could exclude the occurrence of ETV6-RUNX1 whereas low expression of TdT and specific expression of CD81 (along with other conditions—see full rules in results section) positively correlates with ETV6-RUNX1 occurrence.

### 4.2. t(v;11q23)/KMT2A

It is already well known that positive expression of ectopic NG2 marker on blast cells in BCP-ALL is strongly associated with different KMT2A gene rearrangements [1,2,3,4,6]. Since the presence of NG2 marker was from the statistical point of view strong enough to serve as a predictive marker of KMT2A rearrangement occurrence, we constructed the decision tree without NG2. Our results show that even without using NG2, the occurrence of KMT2A rearrangements can be indicated by the lack of CD10, CD34 and low TdT expression. This phenotype corresponds to pro-B-ALL subtype by EGIL (European Group for the Immunological Characterization of Leukemias). Association of the pro-B-ALL phenotype with KMT2A rearrangements was already confirmed in the past by other research groups [1,6]. Added value in KMT2A rearrangements prediction can also be assigned to low (or lack of) CD24 expression which is concordant to the findings of Hrusak et al. [3]. Tsagarakis et al. [2] proposed phenotype of blast cells: CD10-,CD34+/-, NG2+, CD15+, CD9+, CD38+, CD123+, CD20-, CD25-, cIgM-, CD27-, CD66c- as a phenotype strongly associated with KMT2A rearrangement. We have also confirmed that high expression of CD15 + CD65 markers and low cIgM can often be associated with this aberration.

### 4.3. Hyperdiploidy

CD123 marker was essential for prediction of hyperdiploidy in our analysis. Additional important markers in our both predictive and explanatory analysis were CD24, CD34, CD10 and CD9. Although there are some reports describing these markers, gradual assessment of expression allowed us to produce results with high specificity and decent odds ratio. CD123 is commonly known to be associated with hyperdiploidy [7,8]. Furthermore, other studies suggest that lack of CD45 [3,29] and high expression of CD66c and CD34 on blast cells is often seen in hyperdiploid BCP-ALL cases. In our results, the overexpression of CD34 stands out and confirms previous findings. We also associate positive CD66c expression and low CD45 with hyperdiploidy.

### 4.4. No Aberration

We have proposed “no aberration” class to test what antigenic patterns can exclude the occurrence of the three aberrations analyzed in this study. The following phenotype: CD123-/NG2-/CD45 low+/CD22 low+/CD81 high+ indicates that none of the studied aberrations is present, which was not reported previously. Notably, high expression level of CD81 is of added value to exclude the common aberrations which is a novelty in this matter. Absence of NG2 and CD123 is not surprising, as these markers are strongly associated with KMT2A and hyperdiploidy, respectively.

### 4.5. t(12;21)/ETV6-RUNX1 vs. Hyperdiploidy

We have observed some overlap between phenotypic features associated with ETV6-RUNX1 and hyperdiploidy. Therefore, we decided to test what parameters could distinguish one aberration from another. The most useful discriminative markers were CD66c and CD123. Even low expression of these two antigens is more frequent in patients without ETV6-RUNX1 but with hyperdiploidy. Another useful marker is CD10—high expression (especially with conditions described above) indicates hyperdiploidy while low positive would be more frequent in patients with ETV6-RUNX1.

## 5. Conclusions

With the use of the cross-validation methodology, we presented the efficiency of detecting aberrations based on the antigen expression level (Table 4). The efficiency values reported in Table 4 are slightly lower than those reported for statistical analysis without independent test data set isolation [2]. However, the values are still high, particularly in Positive Predictive Value and Negative Predictive Value.

Furthermore, by using advanced methods of data analysis, we have expanded and specified the meaning of particular expression levels of these markers by using gradual expression assessment rather than percentage positivity or qualitative annotations (positive, negative, dim, heterogeneous etc.). In a descriptive analysis we have shown that a multidimensional analysis based on rule induction makes it possible to identify—within the groups of patients with (or without) a given aberration—subgroups of patients with significantly higher chance of a given aberration occurrence (no occurrence) (Rules 1–10). Particularly, it was demonstrated for the so called positive rules (1, 3, 5, 7, 9) (Figure 4) how adding successive conditions increases this chance.

Finding immunophenotype-genotype correlations in pediatric ALL patients would be useful to predict the prognosis at the initial stages of diagnostics.

We have also found new associations, i.e., high levels of CD81 might be useful for exclusion of the common aberrations (ETV-RUNX1 and KMT2A gene rearrangements as well as numerical aberration—hyperdiploidy) occurrence which was not reported before.

Our research confirms previous findings about antigens expression associated with particular genetic aberrations, such as CD10, CD22, CD24, CD34, CD66c, CD45, NG2, CD123. A potential constraint of the study could be the lack of cell viability staining. However, in our opinion this might have only a minor influence on the MFI of particular antigens.

Additionally, we created (and made available) an application for users to predict the occurrence of aberrations studied herein, based on the expression of multiple markers evaluated in a normalised fashion by flow cytometry. The application allows:to recalculate the antigen expression level to the nMFI scale;to make a decision about the aberrations occurrence based on the antigen expression (it is possible to classify a single example or a set of examples);to explain the reasons of classifier-made decisions for each of the considered decision problems.

The full utility of the application requires following strict laboratory sample handling procedures and flow cytometer settings (i.e., EuroFlow or other standardized protocols). This applies to both patient and normal bone marrow samples, which are necessary for construction of normalized marker expression scale, as described in methods section.

## Figures and Tables

**Figure 1 jcm-11-02281-f001:**
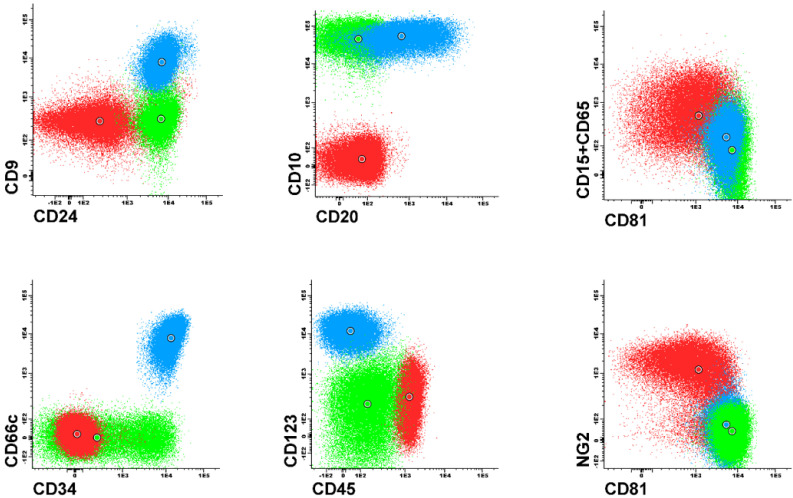
Example of 2–dimensional visualisation illustrating good separation of patients with different genetic aberrations based on phenotypic features.

**Figure 2 jcm-11-02281-f002:**
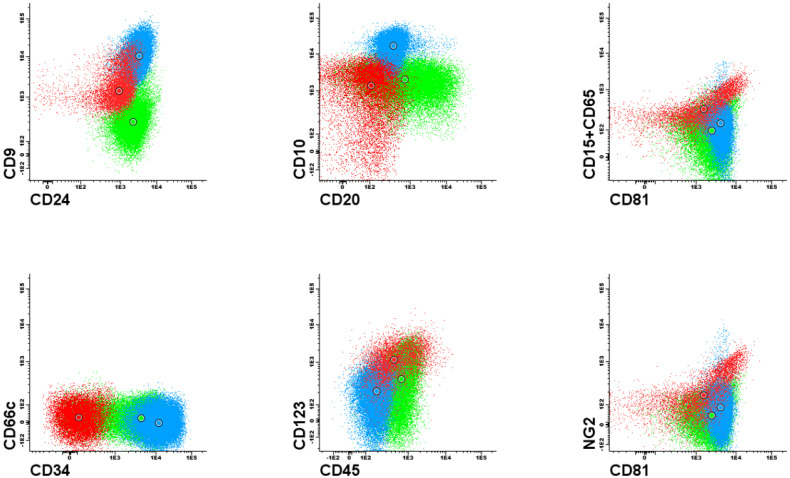
Example of 2–dimensional visualisation illustrating poor separation of patients with different genetic aberrations based on phenotypic features.

**Figure 3 jcm-11-02281-f003:**
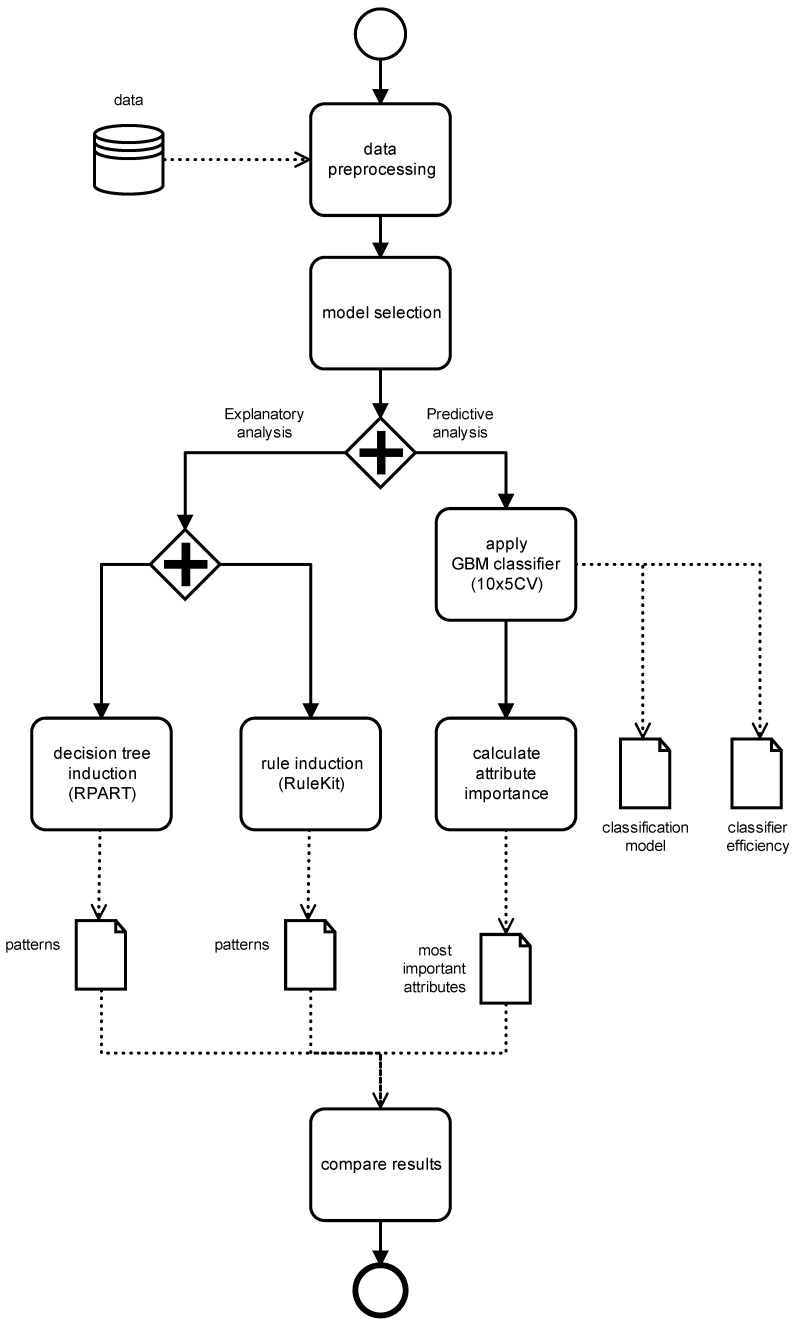
Descriptive and predictive analysis workflow.

**Figure 4 jcm-11-02281-f004:**
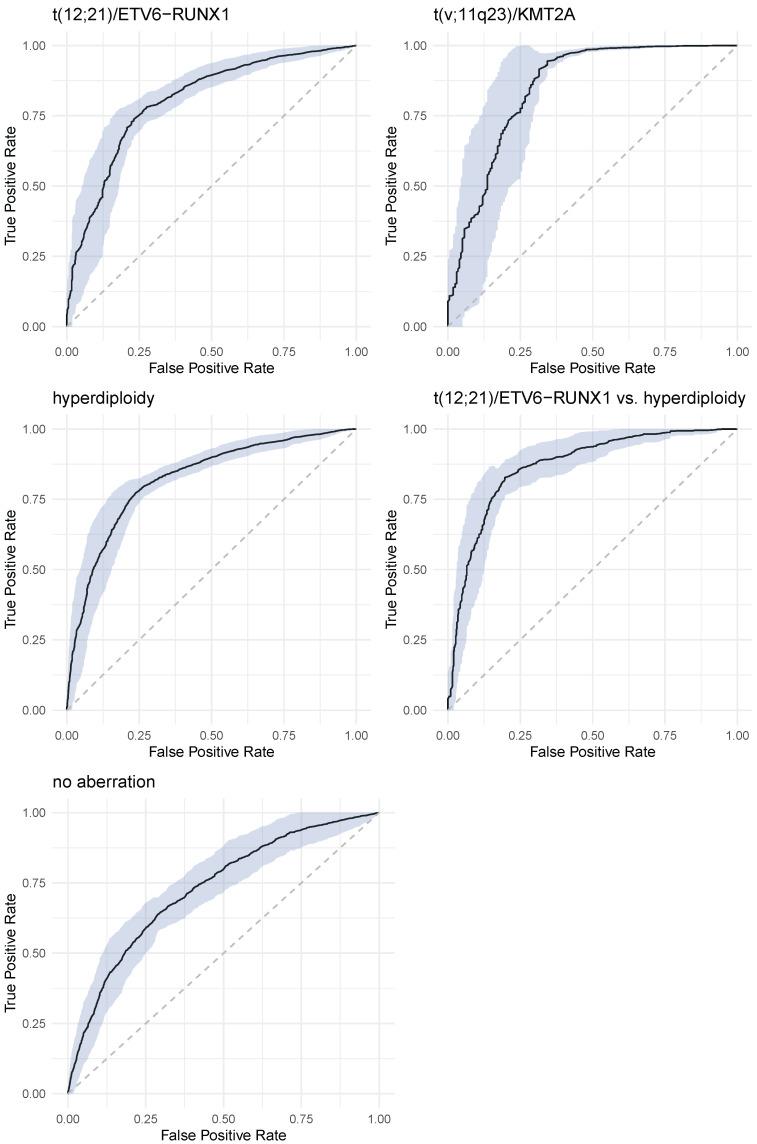
Receiver operating characteristic analysis for t(12;21)/ETV6-RUNX1 and t(x;11q23)/KMT2A prediction.

**Figure 5 jcm-11-02281-f005:**
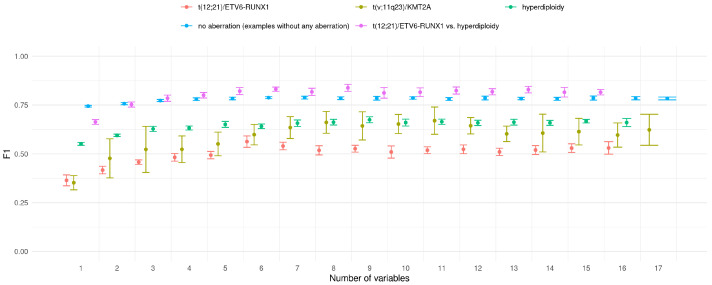
F1 values for classifiers with increasing number of features (mean +/− standard deviation). Each successive classifier uses one more feature than the previous one (i.e., classifier 5 uses all the feature of classifier 4 and the feature ranked 5th in the feature importance ranking).

**Figure 6 jcm-11-02281-f006:**
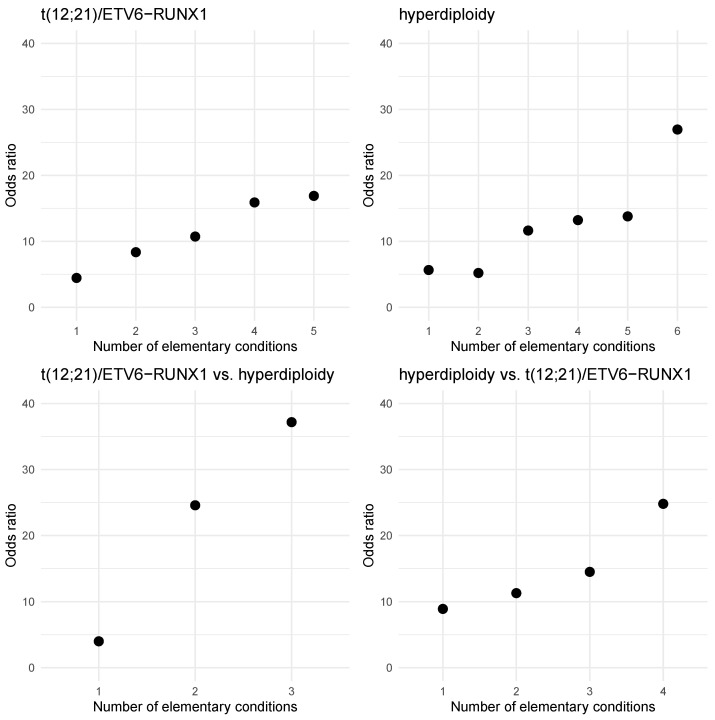
Odds ratio values for the for the premise of the positive rules with successive elementary conditions added.

**Table 1 jcm-11-02281-t001:** Descriptive statistics of antigen expression.

Attribute	Min.	Median	Max.	Attribute	Min.	Median	Max.
NG2	0	0	1	CD81	0	3	25
CD45	0	2	10	cyIgM	1	2	20
CD34	0	4	51	CD123	0	0	9
CD10	0	27	164	CD22	0	2	14
CD20	0	0	15	CD66c	0	1	87
CD38	0	2	30	CD33	0	0	2
CD24	0	22	151	CD9	0	3	58
TdT	1	4	29	CD13	0	2	37
CD15 + CD65	0	0	3				

**Table 2 jcm-11-02281-t002:** Analysed data sets characteristics—decision class distribution.

Decision Problem	Primary Class (1)	sec. Class (0)	Examples
t(12;21)/ETV6-RUNX1	52	422	474
t(x;11q23)/KMT2A	17	588	695
hyperdiploidy	124	378	502
no aberration	257	190	447
t(12;21) vs. hyperdip.	49	121	170

**Table 3 jcm-11-02281-t003:** Classification efficiency (%).

Decision Problem	Sensitivity	Specificity	PPV	NPV
t(12;21)/ETV6-RUNX1	68.1 (5.7)	88.4 (3.1)	42.8 (4,7)	95.8 (6.2)
t(v;11q23)/KMT2A	59.4 (7.5)	98.9 (1.1)	66.0 (15.4)	98.8 (0.2)
hyperdiploidy	78.5 (4.1)	81.4 (2.3)	58.1 (1.8)	92.1 (1.2)
no aberration	92.6 (1.7)	41.4 (4.8)	68.2 (1.3)	80.1 (2.2)
t(12;21) vs. hyperdip.	81.4 (5.2)	91.6 (4.3)	81.5 (6.7)	92.1 (1.9)

**Table 4 jcm-11-02281-t004:** Classification efficiency of RPART algorithm (%).

Decision Problem	Sensitivity	Specificity	PPV	NPV
t(12;21)/ETV6-RUNX1	73	99	95	97
t(v;11q23)/KMT2A	71	100	100	99
hyperdiploidy	86	96	86	96
no abberation	91	89	92	88
t(12;21) vs. hyperdip.	81	98	95	92

## Data Availability

An application to predict the occurrence of selected aberrations: https://flow2abrr.shinyapps.io/appl/. The data sets investigated in the study: https://flow2abrr.shinyapps.io/appl/ and http://adaa.polsl.pl/index.php/datasets-software/ (accessed on 13 April 2022).

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
