# Peer review of "Machine Learning Based Analysis of Relations between Antigen Expression and Genetic Aberrations in Childhood B-Cell Precursor Acute Lymphoblastic Leukaemia"

_jcm, 2022, doi:10.3390/jcm11092281_

Round 1

Reviewer 1 Report

In this study, Kulis J et al. had the aim to evaluate specific genetic aberrations by finding specific multiple antigen expression patterns with FC immunophenotyping. The authors concluded that the t(12;21)/ETV6-RUNX1 aberration occurs more often when blasts present high expression of CD10, CD38, low CD34, CD45 and specific low expression of CD81. Whereas the t(v;11q23)/KMT2A is associated with positive NG2 expression and low CD10, CD34, TdT and CD24.

Moreover, hyperdiploidy is associated with CD123, CD66c and CD34 expression on blast cells. And a high expression of CD81, low expression of CD45, CD22 and lack of CD123 and NG2 indicates that none of the studied aberrations is present. Thus, the authors suggest that they present an original approach of finding multiple (multi-dimensional) antigen expression patterns to identify possible genetic abnormalities in BCP-ALL cases.

In my viewpoint, this is a fantastic manuscript; it provides beneficial information to identify better tools for diagnostic of BCP-ALL. However, it requires changes.

Mayor comments

  1. Normally to develop a flow cytometry assay, viability staining is used. How do you explain the absence of the use of one of them? Because even if the EuroFlow BCP-ALL diagnostic panel did not include it, maybe you should consider it when you designed your panel. It is to be entirely sure that the molecule’s profile is associated with live cells.
  2. I suggest providing a general description of your study population (age, gender, diagnostic time, among others). It could be helpful to propose using the technique in a general or specific population (of course, inside of a BCP-ALL population).

Minor comments

  1. Line 22 and 23, you duplicated the word “cells” I think that one should be removed.
  2. I suggest that lines 55-60 are included as the last section (after conclusions) called “Limitation of the study”, and maybe you can discuss these limitations a little more.

Author Response

Major comments

  1. Normally to develop a flow cytometry assay, viability staining is used. How do you explain the absence of the use of one of them? Because even if the EuroFlow BCP-ALL diagnostic panel did not include it, maybe you should consider it when you designed your panel. It is to be entirely sure that the molecule’s profile is associated with live cells.

The authors thank the reviewer for recognizing the importance of viability staining. Unfortunately, because the study was retrospective (the study group was stained and diagnosed in years 2008-2013) it is now unfortunately not possible to include any viability staining. Cell analysis and blasts gating followed the exclusion of doublets and debris based on the scatter parameters which was assumed sufficient in diagnostic samples. However, inclusion of viability dye would in fact help exclude any unspecific binding of monoclonal antibodies. Appropriate statements were added in Materials and methods and Conclusion sections of the revised manuscript.

  1. I suggest providing a general description of your study population (age, gender, diagnostic time, among others). It could be helpful to propose using the technique in a general or specific population (of course, inside of a BCP-ALL population).

The authors thank for the comment on description of the study population. New information on the structure of the study group was included in Materials and methods section of the revised manuscript. Because of relative high complexity of already obtained results (e.g. the decision trees and rules) on general population of childhood BCP-ALL patients, the authors decided not to subdivide the study group to prevent oversizing the results.

Minor comments

  1. Line 22 and 23, you duplicated the word “cells” I think that one should be removed.

The authors thank for this remark. The typing error was corrected.

  1. I suggest that lines 55-60 are included as the last section (after conclusions) called “Limitation of the study”, and maybe you can discuss these limitations a little more.

The authors thank for this remark. However, this paragraph was meant to point out the limitations of other studies in this area and to contrast our innovative approach exploiting quantitative antigen expression measure and advanced multidimensional data analysis, including cross-validation of the data and releasing the application for other researchers. This paragraph was rephrased for better clarity in the revised version of the manuscript.

Reviewer 2 Report

It is a very interesting and valuable work regarding the information provided by the immunophenotype regarding the genetic aberrations in childhood BCP-ALL.

The importance given to the intensity of each marker is very interesting.

I don't know if this extensive description of the mathematical models should be in the actual paper or in the supplementary information. The editor should decide about it.

The authors should explain if the average flow cytometry user could rely on the usual bright, moderate or weak expressions of the antigens or their combinations to provide information about the genetic aberrations or should use the whole machine based analysis to use their valuable conclusions.

Small typing mistakes should be corrected.

Author Response

It is a very interesting and valuable work regarding the information provided by the immunophenotype regarding the genetic aberrations in childhood BCP-ALL.

The importance given to the intensity of each marker is very interesting.

The authors thank the reviewer for recognizing the importance of this research.

  1. I don't know if this extensive description of the mathematical models should be in the actual paper or in the supplementary information. The editor should decide about it.

The authors decided to include the extensive description of the mathematical models so they could be easily available for readers from such areas as mathematics, computational sciences or machine learning techniques. We would like to keep this description in the manuscript. Supplementary materials contain more detailed information on the used methodology.

  1. The authors should explain if the average flow cytometry user could rely on the usual bright, moderate or weak expressions of the antigens or their combinations to provide information about the genetic aberrations or should use the whole machine based analysis to use their valuable conclusions

The authors thank for the comment on the usage of usual antigen expression notation. Despite frequent usage of the classical notation in the laboratory practice, the use of these terms would not meet the aim of the study. Gradual antigen expression scale enabled to investigate possible genetic abnormalities associated with quantified level of expression of several markers simultaneously while classical notation is only qualitative. Additionally, the introduction of gradual/numerical expression scale in this study was necessary to perform all the calculations. Additional explanation was included in the Materials and methods section of the revised version of the manuscript.

  1. Small typing mistakes should be corrected.

The authors thank for this remark. The typing errors were corrected.